# T Cell Responses in Pregnant Women Who Received mRNA-Based Vaccination to Prevent COVID-19 Revealed Unknown Exposure to the Natural Infection and Numerous SARS-CoV-2-Specific CD4- CD8- Double Negative T Cells and Regulatory T Cells

**DOI:** 10.3390/ijms25042031

**Published:** 2024-02-07

**Authors:** Christina D. Chambers, Jaeyoon Song, Ricardo da Silva Antunes, Alessandro Sette, Alessandra Franco

**Affiliations:** 1Department of Pediatrics, School of Medicine, University of California San Diego, La Jolla, CA 92093, USA; chchambers@health.ucsd.edu (C.D.C.); jac037@health.ucsd.edu (J.S.); 2Center for Vaccine Innovation, La Jolla Institute for Immunology (LJI), La Jolla, CA 92037, USAalex@lji.org (A.S.); 3Department of Medicine, Division of Infectious Diseases and Global Public Health, University of California San Diego, La Jolla, CA 92093, USA

**Keywords:** SARS-CoV-2 vaccination in pregnancy, natural COVID-19 infection, CD4- CD8- double negative (DN) T cells, regulatory T cells, immune regulation

## Abstract

We studied T-cell responses to SARS-CoV-2 in 19 pregnant subjects at different gestational weeks who received three doses of mRNA-based vaccination to prevent COVID-19. SARS-CoV-2 peptide pools were used for T-cell recognition studies: peptides were 15 amino acids long and had previously been defined in COVID-19-convalescent subjects. T-cell activation was evaluated with the AIM assay. Most subjects showed coordinated, spike-specific CD4+ and CD8+ T-cell responses and the development of T cell memory. Non-spike-specific T cells in subjects who were not aware of previous COVID-19 infection suggested a prior undetected, asymptomatic infection. CD4- CD8- double negative (DN) T cells were numerous, of which a percentage was specific for SARS-CoV-2 spike peptides. Regulatory T cells (Treg), both spike- and non-spike-specific, were also greatly expanded. Two Treg populations were defined: a population differentiated from naïve T cells, and pTreg, reverting from pro-inflammatory T cells. The Treg cells expressed CCR6, suggesting homing to the endometrium and vaginal epithelial cells. The pregnant women responded to SARS-CoV-2 vaccination. Asymptomatic COVID-19 was revealed by the T cell response to the non-spike peptides. The numerous DN T cells and Treg pointed our attention to new aspects of the adaptive immune response in vaccine recipients.

## 1. Introduction

Early data at the beginning of the SARS-CoV-2 pandemic suggested that pregnant women were more susceptible to severe COVID-19 and were at greater risk of adverse outcomes such as pre-term delivery [1,2].

The immunity present at the maternal–fetal interface is unique during pregnancy to allow T-cell tolerance of the fetal allograft. T-cell tolerance allows for successful implantation as well as maintenance of a pregnancy to term. The effector immune cells that reside at the interface between the placenta and the uterus are enriched or excluded from the decidua, depending upon their regulatory or inflammatory function [3,4,5,6]. Dendritic cells (DC), the antigen-presenting cells (APC) that activate T cells via the presentation of peptides through the human leucocyte antigen (HLA), are not numerous in the decidua. The maternal adaptive cellular immunity that responds to pathogens is restrained from bi-directional trafficking across the placenta. The decidua is rich in unique lineages of suppressive myeloid DC that can be found in circulation: tolerogenic dendritic cells (tmDC), first discovered in young children and found to be numerous in COVID-19-convalescent pregnant women, are defined by the expression of CD4, CD14, the immunoglobulin-like transcript (ILT)-4 with the canonical markers CD11^c^ and CD11^b^ [7,8] and ILT-4+, and HLA-G+ DC, named DC-10, which is very important in orchestrating the early interaction between maternal immunity and the trophoblast [9,10]. Suppressive DC prime Treg, which are the ultimate key for immune homeostasis at the maternal–fetal interface.

Functionally, Treg restore immune tolerance to *self* and the allo-antigens controlling the milieu where antigen presentation to the T cell occurs, regulating not only inflammation at large but also the differentiation of naïve T cells [11].

During the COVID-19 pandemic, it has been reported that the SARS-CoV-2 spike-specific regulatory T cells (Treg) determined the severity of the disease [12], which was correlated with vaccine potency in healthy donors [13]. Treg expansion has been considered as a possible tool to mitigate the severity of the symptoms [14]. During SARS-CoV-2 infection in pregnancy, the balance between Th17 and pTreg plays a role in the pregnancy outcome [15]. Our laboratory reported the expansion of SARS-CoV-2 spike-specific Treg in pregnant women convalescing from COVID-19 who did not experience adverse reactions or pregnancy complications after the infection, irrespective of pre-existing conditions [8].

In vaccine recipients, the humoral and cellular immune response and its memory development with Moderna mRNA 1273, Pfizer/BioNTechBNT162b2, Jansen Ad26.COV2.S, and Novavax NVX-CoV2373 have been widely described [16,17], suggesting the development of B cell and T cell memory. Work in progress is addressing the functional protective role of antibodies, depending upon fine specificities and affinities, along with the fine specificity of the T cell repertoire, by characterizing immunodominant epitopes.

COVID-19 infection in pregnancy has been associated with an increased risk of poor outcomes, including pre-term births and stillbirth: SARS-CoV-2 affects the placenta in many ways, with a physiopathology that seems to vary between different viral variants reviewed in, although vaccination did not prevent the occurrence of further SARS-CoV-2 infections [18].

Here, we have sought to characterize T cell lineages and their memory phenotype in response to mRNA vaccination in pregnant women who had or had not had a previous COVID-19 infection.

## 2. Results

### 2.1. Characterization of SARS-CoV-2 Spike-Specific CD4+ and CD8+ T Cells and Their Memory Phenotype in Vaccinated Pregnant Women

PBMC were isolated with Ficoll-Hypaque from heparin-treated blood taken from 19 pregnant subjects, studied at different times in pregnancy, who received 3 doses of mRNA-based vaccination within 2 months from sample collection. The race, age, timing in pregnancy, vaccine-related history, and number of deliveries can be found in Table 1. Subjects 2, 5, 10, 16, 17, 18, and 19 reported a COVID-19 infection before pregnancy.

This approach has been extensively validated in acute and convalescent samples [19,20,21,22].

CD4+ T cell responses were evaluated by measuring the expression of two co-stimulatory molecules, OX-40 and 4-1BB; CD8+ T cell responses were evaluated by measuring the expression of 4-1BB and CD69. T cell memory populations were defined as terminally differentiated effector memory (T_EMRA_, CD45RA+ CCR7-), effector memory (T_EM_, CD45RA- CCR7-), and central memory (T_CM_, CD45RA- CCR7+). Data are presented as stimulation index (SI) values, defined as the percentage of T cells responding to peptides, divided by the un-stimulated control signal.

Seventeen of the 19 subjects showed a good CD4+ Th1 response to the spike peptide pool, except for subjects 5 and 11, who did not show a significant response (Figure 1A, upper panel). Interestingly, a low CD4+ Th response was seen in subject 5, who had had COVID-19 before pregnancy, in addition to 3 vaccine doses. Seventeen of the 19 subjects showed a CD8+ CTL response to the spike peptide pool. Subjects 4, 5, 10, and 15 had a low but detectable CD8+ CTL response. Only 2 subjects, 2 and 18, did not show a CTL response, regardless of any previous COVID-19 infection (Figure 1A, lower panel). These differences did not correlate with the timing in pregnancy, nor with exposure to SARS-CoV-2 in addition to vaccination. The percentage of spike-specific activated CD4+ and CD8+ T cells versus the un-stimulated control in vaccinated subjects who did not report a previous COVID-19 infection, and in vaccinated subjects who reported a previous COVID-19 infection, is shown in Appendix A.

The expression of the chemokine receptor CCR6, which determines homing to the endovascular endothelial cells, lungs, gut, neurons, myometrium, and vaginal epithelial cells was also evaluated on AIM+ T cells, and it was identified in a large percentage of both CD4+ and CD8+ T cells (Median = 16.3 and 9.6, respectively; Figure 1B, left panel).

Memory effector T cells that were terminally differentiated (T_EMRA_), effector memory T cells (T_EM_), and central memory T cells (T_CM_) were analyzed in CD4+ and CD8+ AIM+ spike-specific T cell lineages (Figure 1B, right panel).

The CD4+ Th T_EM_ were very numerous and represented the largest spike-specific T cell memory population in circulation (Median 60.2). CD4+ Th also appeared as T_CM_ in a significant percentage (Median 20.8), but not T_EMRA_ (Median 1.1). In contrast, the memory phenotype of CD8+ spike-specific CTL showed a large percentage of T_EMRA_ (Median 56.1). T_EM_ CD8+ were numerous in some subjects (Median 25.9), while the T_CM_ were undetectable (Median 0) (Figure 1B, right panel). Interestingly, previous COVID-19 infection (color-coded in red in Figure 1, panel B) in the vaccine recipients did not show a significant impact on the development of T-cell memory.

### 2.2. Numerous CD4- CD8- Double Negative T Cells with a Mature T Cell Receptor (TcR) Recognized Spike Peptides

In this study population, when we gated CD3+ T cells to analyze SARS-CoV-2-specific CD4+ Th and CD8+ CTL, we found many CD4- CD8- double negative (DN) T cells. DN T cells had a functional T cell receptor (TcR) that responded to anti-CD3/CD28 stimulation (Figure 2).

In 17 of the 19 subjects, a percentage of CD4- CD8- DN T cells were specific for the spike SARS-CoV-2 peptide pool; in some subjects, they were very numerous, being positive for the AIM markers tailored for CD8+ T cell responses, namely, 4-1BB and CD69 (Figure 2). The expansion of DN T cells, with a functional TcR and a percentage responding to the spike peptides, is not unique in vaccinated pregnant women: DN T were significantly expanded in both female and male healthy vaccinated controls (Appendix A).

### 2.3. SARS-CoV-2 Non-Spike-Specific T Cells Were Found in Some Vaccinated Pregnant Subjects Who Were Not Aware of Previous COVID-19 Infection

Most recently, it has been determined that the detection of a T cell response to SARS-CoV-2 non-spike regions in healthy vaccinated donors indicates exposure to COVID-19 infection [23].

We addressed possible asymptomatic exposure to COVID-19 in our cohort by testing the T cell responses to 2 peptide pools spanning the non-spike regions of SARS-CoV-2 [23], one containing 69 peptides (pool 1), the one recognized by the general adult population, and a second pool containing 215 peptides (pool 2). T cell reactivity was evaluated by AIM assay, in parallel, and as previously described for the evaluation of spike-specific T cell responses.

Among the subjects who reported COVID-19, namely, 2, 5, 10, 16, 17, 18, and 19 (Table 1), we found different patterns of T cell recognition of the non-spike peptides. Subjects 16, 17, and 18 showed CD4+ Th expansion in response to non-spike peptide epitopes, but not to CD8+ T cells; subject 5 had only a detectable CD8+ T cell response, and subject 2 did not have non-spike-specific T cells in circulation (Figure 3A).

Interestingly, the study showed that subjects 1, 3, 4, 8, 9, 12, and 14 did not know about their COVID-19 infection before or during pregnancy, but had a T cell response to the non-spike peptide epitopes. Subjects 1, 12, and 14 showed a coordinated CD4+ Th and CD8+ CTL response to the non-spike peptide pools, and subjects 3, 4, 8, and 9 only showed a CD4+ T helper response (Figure 3A). Pool 1 (69 amino acids) and pool 2 (215 amino acids) were found to be immunogenic without significant differences.

CCR6 was expressed in a significant fraction of CD4+ Th cells: pool 1 Median 18.8; pool 2 Median 20 and CD8+ T cells, pool 1 Median 10, and pool 2 Median 29.1, as previously observed for the spike-specific T cell population (Figure 3A, right panels). The results indicated a similar tissue distribution as the spike-specific T cells. The expression of CCR6 on CD8+ T cells was a unique feature in vaccinated pregnant women compared to adult vaccinated controls, where CCR6 was found only on CD4+ T cells [13].

The non-spike-specific T cell memory phenotype suggested that CD4+ Th T_EM_ were numerous and represented the largest non-spike-specific T cell memory population in circulation: pool 1 Median 53.5; pool 2 Median 49.4; CD4+ Th were T_CM_ in a significant percentage also pool 1 Median 21.9, pool 2 Median 27.1 (Figure 3B). CD8+ memory T cells were less numerous than spike-specific T cells: T_EMRA_ were enumerated in 4 subjects, pool 1 Median 37.6 and pool 2 Median 35.9; T_EM_ CD8+ non-spike-specific cells were detectable in 3 subjects: pool 1 Median 0, pool 2 Median 17.1 and T_CM_ only in 1 subject (Figure 3C).

Taken together, the results suggested that several vaccinated pregnant women had had COVID-19 previously, but they were not aware of it, and that the natural infection, in addition to vaccination, did not boost the development of T cell memory in this cohort.

### 2.4. SARS-CoV-2 Spike and Non-Spike-Specific Treg Were Numerous in Vaccinated Pregnant Women

When we enumerated spike-specific and non-spike-specific CD4+ CD25^high^ Treg, we found specific T cell populations in 18 of the 19 subjects and in 11 of the 19 subjects, respectively (Figure 4, upper and lower panel A). An enumeration of Treg in the unstimulated control can be found in Appendix A.

A percentage of the Treg expressed the chemokine receptor CCR6: spike-specific Median 8.4, non-spike-specific Medians 15.2 and 11.6 for pool 1 and pool 2 respectively (Figure 4, panel B), suggesting homing to the vascular endothelium, lung, gut, neurons, and most importantly in pregnancy, the endometrium and vaginal epithelial cells. Treg developed memory in vaccinated pregnant women, as we recently reported in healthy vaccinated subjects [13]. Treg T_EM_ were numerous, spike-specific Median 60.1, and non-spike-specific Median 55.3 (pool 1) and 61.1 (pool 2); a great percentage were also T_CM_, spike-specific Median 21, non-spike-specific Median 22.9 (pool 1), and 20.6 (pool 2) (Figure 4, panel C).

An assay of IL-10 in culture supernatants of PBMC, stimulated for 24 h with spike and non-spike peptides, further supported the Treg response (Figure 4, panel D).

Next, we considered the ontogeny of those Treg that so efficiently expand after mRNA-based vaccination. Peripherally-induced Treg (pTreg), which revert from antigen-specific pro-inflammatory T cells, are defined by the expression of the IL-7 receptor (IL-7R) [24,25] and return to a CD45RA+ phenotype (reviewed in [11,26]). Our previous work in healthy adult vaccine recipients suggested that SARS-CoV-2-specific Treg have a unique phenotype, comparable to natural regulatory T cells (nTreg) [13].

We further characterized Treg by studying specific markers via flow cytometry. The markers included FOXP3 intracellular, CTLA-4, and PD-1, a hallmark of Treg, together with CD45RA and IL-7R, which define pTreg. 

To better understand the possible conversion of pro-inflammatory T cells to pTreg over time, we explored Treg phenotypes at different time points when in culture.

PBMC cultures from subjects 13, 14, and 16 were stimulated with SARS-CoV-2 spike and non-spike peptide pools for 48 h. The role of IL-2 concentration in influencing the Treg phenotype was explored in PBMC cultures from donor 14. The expression of PD-1 is plastic on Treg and is modulated by IL-2, determining the extent of suppression by Treg of pro-inflammatory T cells during infections [27].

The choice of the subjects was unbiased, depending solely upon cell numbers after PBMC separation. The cells in culture were collected, washed, and studied for the expression of specific Treg markers at 48 h after and 4 days after stimulation with peptides, respectively.

The results were unexpected: when we enumerated the percentage of spike-specific and non-spike-specific Treg versus the un-stimulated control, we found numerous FOXP3+ CD4+ CD25high Treg expressing CTLA-4 in a small percentage, but not PD-1. Spike- and non-spike-specific Treg were IL7R- CD45RA- in terms of natural Treg (nTreg) (Figure 5, panel A).

To address the possible dependency on IL-2 in the environment in determining the expression of PD-1, as previously observed in spike-specific Treg from healthy donors [13], we expanded Treg from subject 14 with two different concentrations of IL-2. Surprisingly, IL-2 did not have any effect on PD-1 expression, unveiling a unique phenotype of SARS-CoV-2 spike-specific Treg in pregnancy (Figure 5, panel B).

However, after 4 days in cultures with peptide antigens in the absence of IL-2, we enumerated a high percentage of Treg IL-7R+ and anti-CD45RA+, showing the development of pTreg over time (Figure 5, panel C).

The percentage of Treg after 48 h of stimulation with SARS-CoV-2 peptides in the presence or absence of IL-2 (Figure 5 panel B) did not change; furthermore, the percentage did not change after a longer period of stimulation (Figure 5, panel C), waving concerns about by-standard Treg expansion due to IL-2 being secreted by pro-inflammatory antigen-specific T cells, stimulated by antigens.

## 3. Discussion

Herein, we studied the T cells’ responses to SARS-CoV-2 peptides, derived from the spike and non-spike proteins found in vaccinated pregnant women who received up to 3 doses of Moderna mRNA 1273 and/or Pfizer/BioNTechBNT162b2. We addressed the extent of the spike-specific T cell expansion and T cell memory development, the possible unknown natural COVID-19 infection (detected by non-spike-specific T-cell responses), and the development of virus-specific Treg, which are important for protecting the fetus in pregnancy. There are many open questions on the elucidation of the mechanisms involved in protecting the fetus from undesirable maternal pro-inflammatory T-cell responses that are capable of crossing the placental barrier. Low numbers and/or functional defects within the innate and adaptive immune regulation in the maternal–fetal interface may lead to premature delivery, stillbirths, and other clinical complications in premature infants. In physiological conditions, during a viral infection, the maternal immune response clears the pathogens, avoiding fetal damage due to pro-inflammatory T cells in the placental interface. One mechanism that prevents T cell activation is the very low number of DC in the decidua, which limits the ability to initiate adaptive immune responses in the draining lymph nodes not only to fetal antigens but also to antigens derived from viruses or pathogens [6]. The delicate balance between a necessary pro-inflammatory systemic maternal immune response and strong immune regulation within the placenta occurs through complex cross-talk between the maternal immune cells and the trophoblast.

In our study, the primary CD4+ Th and CD8+ T cell responses to SARS-CoV-2 mRNA-based vaccines in pregnancy were comparable to the T cell responses that we reported in healthy vaccinated donors [13]. However, we did find differences within the phenotype and memory development of spike-specific CD8+ T cells. The expression of the chemokine’s receptor CCR6+ on CD8+ T cells was unique in vaccinated pregnant women: our previous work in vaccinated healthy donors suggested CCR6 expression occurs only in CD4+ Th and Treg but not in CD8+ T cells [13], indicating different CTL homing in pregnancy. CCR6 engages CCL20 and is expressed not only on vascular endothelial cells, lung epithelial cells, and the gut and neurons but also myometrium and vaginal epithelial cells. The expression of CCR6 on CD8+ T cells allows T cell trafficking to the uterus and vagina, possibly representing an additional protection for viral placental infections. The CD8+ T cell memory repertoire was also different in vaccinated pregnant women, showing the development of T_EM_, which are numerous in vaccinated pregnant women but not in healthy vaccinated adults or COVID-19-convalescent subjects; CD8+ T cells were T_EMRA_ [13,28], further insuring potent, rapid, and long-lasting protection from possible new SARS-CoV-2 placental infections.

When we tested T cell responses to non-spike peptides, we found that a few of the subjects who had had COVID-19 did not show a significant anti-spike T cell response and T cell memory development, suggesting that the non-spike region of the SARS-CoV-2 virus is less immunogenic for T cell recognition. However, several subjects were responding to the two non-spike peptide pools, although they were not aware of any previous SARS-CoV-2 infection, as previously reported [23]. Non-spike-specific memory T cells expanded as spike-specific memory T cells with a similar phenotype, confirming T_EM_ within the CD8+ memory populations.

There is much data, and speculation, on the consequences of COVID-19 infection in pregnancy and its relevance to the delivery outcome and the development of the fetus. In this study, 6 subjects had a significant T cell response to the non-spike peptide epitopes, unveiling a not previously diagnosed exposure to SARS-CoV-2. It was not possible to determine if the infection occurred before or during pregnancy. 

A puzzling result was the large expansion of CD4- CD8- DN T cells, which were fully functional and comprised a good percentage of SARS-CoV-2 spike-specific cells. DN have been historically described as immature T cells in the thymus, which then develop into CD4+ CD8+ double positive (DP) cells, prior to full maturation to CD4+ or CD8+, depending upon an encounter with molecules of the major histocompatibility complex (MHC) [29]. 

Our work in multisystem inflammatory syndrome in children (MIS-C), an acute inflammation that occurs 5–6 weeks after COVID-19 infection, reported numerous DN T cells that responded to SARS-CoV-2 peptide antigens, expressing after the activation of both CD4+ and CD8+ T cell markers [30]. Numerous DN T cells were SARS-CoV-2-specific in MIS-C, but not all: we discussed, then, the young age of the study population and that possible DN recognition of specificities for co-infecting infectious agent(s) may be the cause(s) of MIS-C, besides SARS-CoV-2. In vaccine recipients, DN T cells were also numerous in healthy controls and are not a hallmark of vaccination in pregnancy. An open question is the fine specificity of the spike- and non-spike-specific DN T cells, which may recognize different peptides from the mature CD8+ T cells. We could not address this question by testing large numbers of peptide epitopes in pools. The TcR affinity and avidity for the HLA/peptide complex could be also different from those seen in mature CD8+ T cells. The expansion of DN T cells may occur within the epitope, spreading due to the pro-inflammatory environment, or because of cross-reactivity or mimicry in the context of strong antigenic stimuli. We propose a role for DN T cells in responding to T cell overstimulation, due to multiple boosts within the SARS-CoV-2 spike mRNA with too-short intervals. On the same line, other vaccine formulations such as influenza, tetanus/diphtheria/pertussis (TDaP), and respiratory syncytial virus (RSV), when administered at the same time or close to SARS-CoV-2 vaccination could lead to DN expansion. In our pregnant cohort, the receipt of multiple vaccinations is consistent with the current standard recommendations for vaccine administration in pregnancy, raising the question of the immune efficacy (and safety) of multiple vaccines being administered at the same time, especially in the third trimester.

The study also revealed the significant expansion of spike- and non-spike-specific Treg, which may protect the fetal compartment, not only by secreting IL-10 and suppressing adaptive and innate pro-inflammatory cell populations but also by expressing CCR6. In this study, Treg were equally expanded in vaccinated subjects and in subjects who had had COVID-19 in addition to vaccination. The importance of Treg in the control of COVID-19 was previously suggested by the evidence that Treg that have been isolated from SARS-CoV-2-infected adults with severe symptomatology have a peculiar phenotype that jeopardizes immune regulatory functions [12]. Several papers further emphasized the importance of Treg in controlling exuberant inflammation in COVID-19 (reviewed in [14]).

After priming with peptide antigens, SARS-CoV-2 spike-specific regulatory T cells do not express IL-7R and CD45RA after two days in culture, suggesting that they do not arise from pro-inflammatory T cells that are repeatedly stimulated, as with classical pTreg, but are rather from naïve T cells in tissues where spike synthesis and antigen presentation to T cells occur. We found spike-specific Treg in vaccinated, non-pregnant healthy donors [13] but, in this pregnant cohort, the expression of CTLA-4 and PD-1 was significantly different. Phenotypically FOXP3+, as classical Treg, the expression of CTLA-4 was very low on spike- and non-spike-specific Treg and PD-1 was not expressed. Moreover, unlike in healthy vaccinated non-pregnant controls, different regimens of IL-2 during Treg priming did not raise the expression of these markers.

The dynamic transcriptional activity and chromatin remodeling of Treg, depending upon the duration of IL-2 signaling, showed that PD-1 expression appears to control Treg effector functions [31]. In the *Toxoplasma gondii* murine model, PD-1/PD-1L interactions modulate the effector function of Treg during infection [27]. A possible explanation is the double role of PD-1, which, depending upon the gradients of expression on immune cells, could serve as a stimulatory molecule. The PD-1 blockade counteracts post-COVID-19 immune abnormalities and stimulates anti-SARS-CoV-2 immune responses [32]. Interestingly, in the absence of IL-2, and with a longer exposure to peptide antigens, the spike-specific pTreg, IL7R+ CD45RA, expand over time, always with a low expression of CTLA-4 and the absence of PD-1. Possibly, virus-specific Treg function predominantly via IL-10 secretion in the placental environment, broadly keeping at bay both innate and adaptive inflammatory responses, involving non-cognate interactions with pro-inflammatory T cells. The expression of CCR6 on Treg suggested crosstalk with tissues, with cognate interactions playing a role in maintaining the physiology of tissues homeostasis. 

This paper has some strengths and limitations. The merit of the study is the report of previously undescribed T-cell lineages after vaccination for COVID-19 protection in pregnancy. Several subjects were convalescing after contracting COVID-19 before pregnancy, giving us the opportunity to evaluate several aspects of the SARS-CoV-2-specific T-cell response, either after vaccination, or after vaccination and prior to COVID-19 infection. This is relevant to the community since, often, datasets on vaccine efficacy that have been derived from healthy donors do not clearly separate those subjects who have had one or multiple COVID-19 infection episodes. 

The study of T cell specificities to the non-spike protein also unveiled a large percentage of previously undetected or asymptomatic COVID-19 infections. 

Our cohort was diverse in terms of age, gestational timing of vaccination, and the receipt of other vaccinations during pregnancy. Besides SARS-CoV-2 protection, some conclusions of this study could be broadly relevant to our understanding of immunity in the maternal–fetal interface. These included the enumeration and characterization of virus-specific Treg, which may play an important role in regulating the impact of maternal pro-inflammatory immune responses in the placenta. These may be stimulated by multiple vaccinations or natural infections that may also lead to the expansion of DN T cells, which are fully functional and respond to SARS-CoV2 spike with CD8+ activation markers, which may be a sign of undesired over-stimulation. 

A substantial limitation of our study was the lack of access to placental tissues to determine T cell distribution and crosstalk between SARS-CoV-2-specific T cells in the villi and the innate immune cells that determine maternal blood communications with the developing embryo. Also, we previously tested the suppressive activity of SARS-CoV-2-specific Treg in healthy vaccinated donors [13]. Here, we showed IL-10 secretion in response to Treg stimulation, but the T cell numbers were limited, rendering unfeasible any co-culture experiments with FACS-sorted pro-inflammatory, virus-specific T cells. 

We were not able to report the extent of SARS-CoV-2-specific T cells in the cord blood. In addition, although the participants consented to tests for both maternal and infant postnatal antibodies to SARS-CoV-2, as well as evidence of infection, at the time of this analysis we did not have this information available, nor did we have information on the extent of antibodies and T cell transfer through lactation.

## 4. Materials and Methods

### 4.1. Study Population

Study population: through the MotherToBaby Pregnancy Studies cohort study at the University of California San Diego, pregnant women residing anywhere in the U.S. or Canada who had been vaccinated for COVID-19 protection consented to providing blood samples following their most recent third vaccination dose during pregnancy with an mRNA-based vaccine. Women with pre-existing conditions have been excluded from the study. The study included the receipt of bivalent formulations of BNT162b2 (Pfizer-BioNTech) and mRNA-1273 (Moderna). The vaccines were based on mRNA encoding the spike protein from the original (ancestral) strain of SARS-CoV-2 and from the B.1.1.529 (Omicron) variants, BA.4 and BA.5 (BA.4/BA.5). Blood was drawn by arrangement with each participant’s closest Quest Laboratory within 2 months of the third vaccine dose in pregnancy. Samples were shipped overnight to the laboratory at the University of California, San Diego. A total of 19 pregnant women were enrolled in the study and provided blood samples. The women ranged in age from 28 to 42 and provided blood samples at gestational weeks ranging from 10 to 34 (Table 1). Blood was drawn from these subjects when the consent form had been signed. Table 1 reflects the gestational time of the study. Of the 19 subjects, 7 reported being aware that they had had a previous COVID-19 infection (Table 1). Ethics approval for the study was provided by the University of California San Diego Institutional Review Board (UCSD IRB #141162). 

Female and male healthy vaccinated controls were studied for spike-specific Treg characterization [13] and were used as controls for the enumeration of CD4- CD8- double negative (DN) T cells, which have previously not been reported.

### 4.2. SARS-CoV-2 Peptides Mega Pool

A SARS-CoV-2 spike peptide pool was used to study the CD4+ and CD8+ T cell responses to SARS-CoV-2 in vaccinated pregnant women. The peptide pools were designed based on the reference genomic sequence of the Wuhan-Hu-1 SARS-CoV-2 isolate (GenBank ID: MN908947), which was described and validated in acute and convalescent SARS-CoV-2-infected patients, as well as in unexposed healthy subjects [19,20]. The SARS-CoV-2 spike peptide pool contains 253 peptides of 15 amino acids long, overlapping by 10 amino acids and spanning the entire spike protein. Non-spike (R) regions were defined from 25 different studies [33]. The selected epitopes were pooled into 2 pools, encompassing those epitopes that have been found to elicit positive responses in three or more donors, n = 69 (pool 1), along with a second pool, n = 215 (pool 2). T-cell responses to these peptides have been used in combination with the spike peptide pool to classify unexposed, naturally infected, vaccinated, and vaccinated convalescing cohorts [23].

The peptides were synthesized as crude material (T.C. laboratories, San Diego, CA, USA), resuspended in DMSO, pooled according to a mega pool design, and re-lyophilized. 

### 4.3. AIM Assay

Peripheral blood mononuclear cells (PBMC) were separated by Ficoll-Hypaque density centrifugation. After counting, 1 × 10^6^ cells were stimulated in 96-well U bottom plates with 1 μg/mL of spike peptide pool and with the non-spike peptide pools. PBMC cultured with 0.1% DMSO, with the same concentration of DMSO (solvent) in the peptide(s) pool-stimulated cultures, served as un-stimulated controls. After 24 h, the cell cultures were harvested, stained with monoclonal antibodies, and analyzed by flow cytometry to study T cell activation, CCR6 expression, and T cell memory phenotypes. To ensure the comprehensive assessment of CD4-spike- and CD8-spike-specific reactivity, the main target of vaccine candidates, with overlapping 15-mers by 10 spanning the entire protein, have been synthesized (253 peptides) based on Wuhan-Hu-1 isolates as a reference (GenBank ID:MN908947). Antigen-specific responses were determined by measuring the expression of T cell markers by flow cytometry by using the activation-induced cell markers (AIM) assay. We measured the co-expression of 4-1BB and OX40, two TNF family member co-stimulatory molecules, up-regulated following T cell receptor signaling on CD4+ T cells, and by measuring the co-expression of 4-1BB and CD69 (adhesion molecule involved in lymphocyte homing and trafficking) on CD8+ T cells [20]. The antibodies included: anti-CD3-AF700 (clone OKT3, mouse IgG2aκ, Biolegend, San Diego, CA, USA), anti-CD4-BV605 (clone RPA-T4, mouse IgG1κ, BD Bioscience, San Diego, CA, USA), anti-CD8-PE/CF594 (clone RPA-T8, mouse IgG1k, BD Bioscience, San Diego, CA, USA), anti-4-1BB-allophycocyanin (clone 4B4-1, mouse IgG1κ, Biolegend, San Diego, CA, USA), anti-OX40-PE/Cy7 (clone Ber-ACT35, mouse IgG1κ, Biolegend, San Diego, CA, USA), anti-CD69-PE (clone FN50, mouse IgG1κ, BD Bioscience, San Diego, CA, USA), anti-CCR6-PerCp/Cy5.5 (clone 11A9, mouse IgG1κ, BD Bioscience, San Diego, CA, USA), anti-CD45RA-APC-H7 (clone HI100, mouse IgG2bk, BD Bioscience, San Diego, CA, USA), and anti-CCR7-FITC (clone G043H7, mouse IgG2aκ, Biolegend, San Diego, CA, USA). Data were recorded on a BD Canto II, San Diego, CA, USA and were analyzed with the FlowJo software version 10 (Tree Star, Ashland, OR, USA). Isotype controls for each antibody were tested and showed no staining. Treg were defined by the co-expression of CD4 (anti-CD4-BV605, clone RPA-T4, mouse IgG1κ, BD Bioscience, San Diego, CA, USA) and CD25^high^, with a mean fluorescence of approximately 10^4^ (anti-CD25-BV421 (clone M-A251, mouse IgG1κ, BD Biosciences, San Diego, CA, USA). The expression of the chemokine receptor CCR6 on AIM+ CD4+ and CD8+ T cells was also analyzed. T_EMRA_, CD45RA+ CCR7-, T_EM_, CD45RA- CCR7-, and T_CM_ CD45RA- CCR7+ were enumerated on AIM+ CD4+ and CD8+ T cells.

### 4.4. Phenotype of SARS-CoV-2 Spike and Non-Spike-Specific Treg

The phenotypical characterization of spike-specific Treg was determined by: surface anti-CD4-PerCp/Cy5.5 (clone RPA-T4, mouse IgG1κ, eBioscience, San Diego, CA, USA), anti-CD25-BV421 (clone M-A251, mouse IgG1κ, BD Biosciences, San Diego, CA, USA), anti-IL-7Rα-FITC (clone eBioRDR5, mouse IgG1κ, eBioscience, San Diego, CA, USA), anti-PD-1-BV605 (clone EH12.2H7, mouse IgG1κ, BioLegend), anti-CTLA-4-PE/Dazzle594 (clone BNI3, mouse IgG2aκ, Biolegend, San Diego, CA, USA), and intracellular anti-FOXP3-PE (clone 259D, mouse IgG1κ, Biolegend, San Diego, CA, USA). Data were recorded on a BD Canto II (BD Bioscience, San Diego, CA, USA) and analyzed with FlowJo software, version 10 (Tree Star, Ashland, OR, USA).

### 4.5. Statistical Analysis

The data were analyzed using Prism software, version 9.0 (GraphPad Software). To compare the percentage of AIM+ T cells in the un-stimulated control and with peptide stimulation, data obtained from the spike peptide megapool-stimulated culture and from un-stimulated controls in the individual cohort were tested using non-parametric paired tests. A *p*-value of ≤0.05 was considered significant.

## Figures and Tables

**Figure 1 ijms-25-02031-f001:**
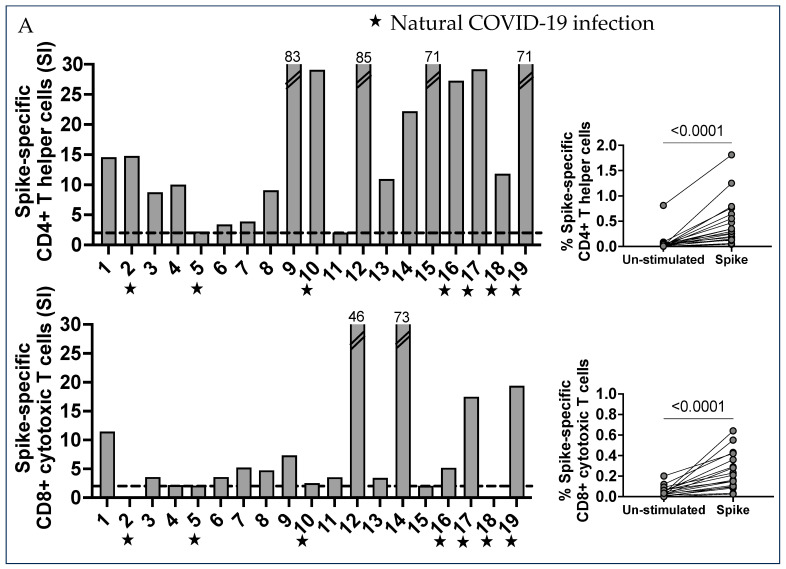
Characterization of SARS-CoV-2 spike-specific CD4+ and CD8+ T cells and the development of T cell memory. (**A**) PBMC were separated from 19 vaccinated pregnant women who received 3 injections of mRNA-based vaccination for COVID-19 protection. PBMC were stimulated in vitro with the spike peptides pool, which includes 253 peptides that are 15 amino acids long, overlapping by 10 amino acids, and spanning the entire spike protein. Then, 24 h after stimulation, the cells were collected and stained with specific monoclonal antibodies to determine the T cell activation state in response to peptide antigens (AIM assay). The percentage of AIM+ CD4+ Th cells from the un-stimulated control and SARS-CoV-2 CD4 spike-stimulated cells, presented as stimulation index (SI) values, were compared using the Wilcoxon matched-pairs signed rank test. The percentage AIM+ T cell increase for the unstimulated samples was a positive T cell response: a stimulation index of 2 (dotted lines) was considered a good T cell response. The percentage of AIM+ CD8+ CTL cells from un-stimulated control and SARS-CoV-2 CD4 S-all-stimulated cells is also presented here, using the stimulation index (SI). Results were compared by the Wilcoxon matched-pairs signed rank test. CCR6 expression was found in a large percentage of SARS-CoV-2 spike-specific CD4+ Th cells in most subjects, and CD8+ CTL cells were also expressing CCR6. (**B**) T cell memory phenotypes. Memory populations were defined by specific markers on gated AIM+ CD4+ Th and CD8+ CTL cells. Each dot shows the percentage of memory populations: terminally differentiated effector T cells (CD45RA+ CCR7- T_EMRA_), effector memory T cells (CD45RA- CCR7- T_EM_), and central memory T cells (CD45RA- CCR7+ T_CM_).

**Figure 2 ijms-25-02031-f002:**
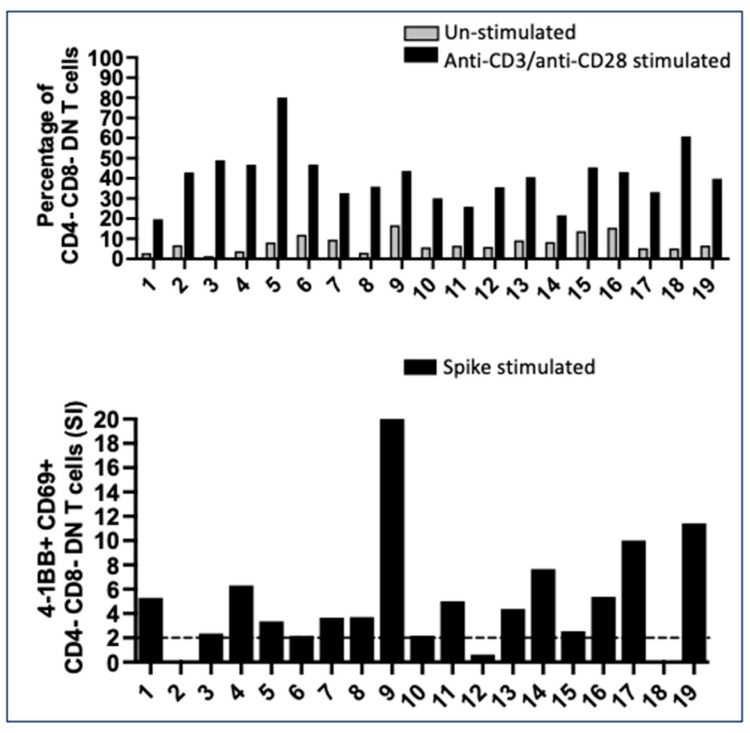
Numerous CD4- CD8- double negative (DN) T cells with a mature T cell receptor (TcR) developed after vaccination. Several CD3+ T cells were CD4- CD8- DN T cells in the 19 pregnant subjects. Upper panel: percentage of DN T cells in un-stimulated PBMC compared to the percentage of DN T cells that responded to mitogenic stimulation with anti-CD3/anti-CD28, proving the maturity of the TcR. Lower panel: CD4- CD8- DN response to spike peptide epitopes, measured as the expression of 4-1BB and CD69, the markers that we used to enumerate CD8+ CTL, suggesting that the function of these DN T cells is related to cytotoxicity. Data are presented as stimulation index (SI) values. Seventeen of 19 subjects had a significant DN spike-specific T-cell response (SI ≥ 2).

**Figure 3 ijms-25-02031-f003:**
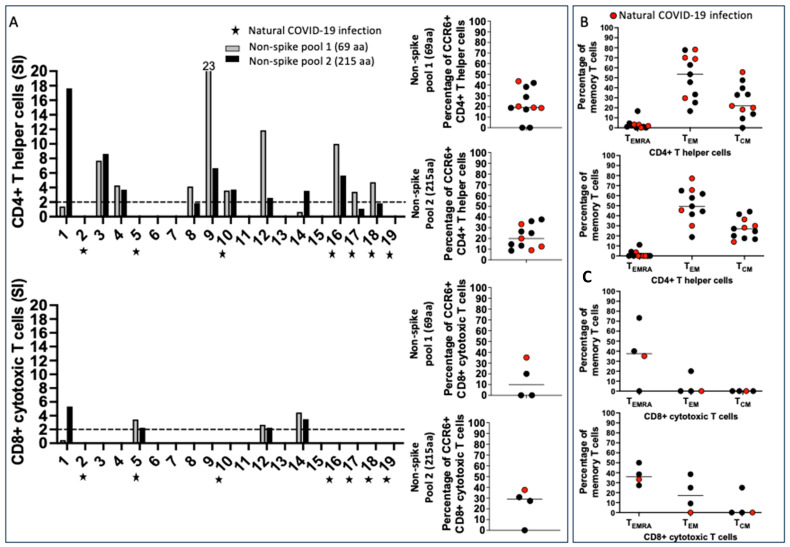
Non-spike-specific T cells were found in some vaccinated pregnant subjects who were not aware of previous asymptomatic COVID-19 infection. The T cell response to non-spike proteins has been evaluated with two peptide pools: pool 1 = 69 amino acids, recognized by most convalescing from COVID-19, and pool 2 = 215 amino acids. PBMC were primed for 24 h prior to studying T cell activation with the AIM assay. (**A**) CD4+ and CD8+ T cell response to non-spike peptides; the percentage of CCR6+ T cells is shown. The two peptide pools have been found to be equally immunogenic in this cohort: all the subjects who had previously been infected, except for subject 5, had a CD4+ Th response. Subject 5 was the only subject who developed an appreciable CD8+ CTL response within the cohort who was aware of previous COVID-19 infection. T cells from several subjects were responding to non-spike peptide epitopes, suggesting that they were unaware of previous asymptomatic COVID-19. CCR6 was found in a considerable percentage of CD4+ and CD8+ non-spike-specific T cells. (**B**) T cell memory phenotypes. Each dot shows the percentage of memory populations: CD4+ Th were mostly T_EM_ and T_CM_, but not T_EMRA_; (**C**) CD8+ CTL were mostly T_EMRA_ and a few T_EM_ and T_CM_, as previously shown for the spike-specific T cell memory repertoire.

**Figure 4 ijms-25-02031-f004:**
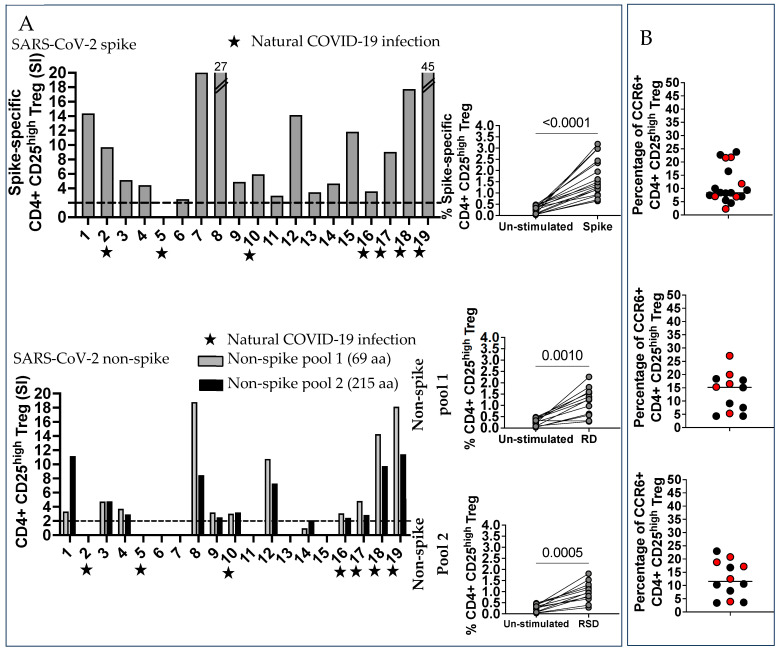
Regulatory T cells (Treg) spike and non-spike-specific were a large percentage of the SARS-CoV-2-specific T cells. (**A**) Regulatory T cells (Treg) were defined as CD4+ CD25^high^. The percentage of CD4+ CD25^high^ Treg from un-stimulated controls and SARS-CoV-2 spike and non-spike-stimulated PBMC were compared by the Wilcoxon matched-pairs signed rank test. Spike-specific Treg were numerous in most subjects (Median 7.49), unless they were non-spike-specific Treg, which were less numerous regardless of previous COVID-19 infection. CCR6 was expressed in a large percentage of spike-specific and non-spike-specific Treg. (**B**) The characterization of memory Treg suggested very numerous T_EM_ spike-specific and non-spike-specific cells and T_CM_ spike and non-spike-specific cells, but not T_EMRA_. (**C**) The dosage of IL-10 in PBMC culture supernatant 24 h after stimulation with peptides and tested by an ELISA confirmed that Treg were numerous in most subjects. (**D**) IL-10 has been measured by ELISA in culture supernatants, 24 h after stimulation with peptides.

**Figure 5 ijms-25-02031-f005:**
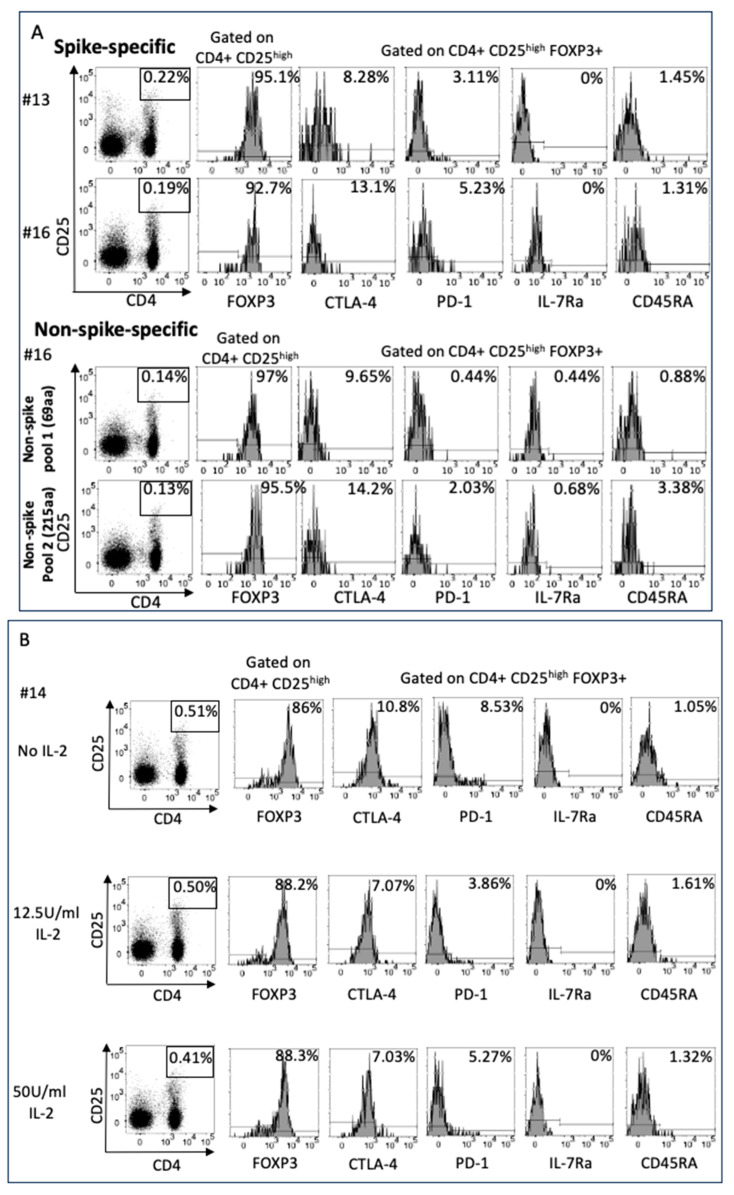
Treg phenotypical characterization and differentiation. (**A**) SARS-CoV-2 spike-specific Treg were phenotypically characterized 48 h after culture with peptides in PBMC of samples from subjects 13 and 16 and of non-spike Treg (specific for pool 1 and pool 2) in samples from subject 16. The markers were chosen to determine Treg ontogeny, possibly reverting from pro-inflammatory T cells (pTreg) or arising from naïve T cells primed in tissues that synthesize spike proteins. Surprisingly, FOXP3+ Treg expressed CTLA-4 in a very small percentage of cells, but not PD-1, and the results were negative for IL-7R and CD45RA, suggesting that they arise from naïve T cells, rather than reverting from pro-inflammatory T cells. (**B**) Treg from subject 14 were also studied to address the role of IL-2 in the expression of PD-1 and were studied under IL-2 feeding (12.5 and 50 U/mL) during T cell priming. Surprisingly, IL-2 does not influence the expression of PD-1 on Treg, which may be a unique feature in pregnancy. (**C**) The PBMC from subject 19 were kept for four days in culture in the absence of IL-2 after T cell priming to determine the possible conversion of pTreg from pro-inflammatory T cells over time. IL-7R + Treg expanded rapidly, with many reverting to double positive IL-7R+CD45RA+ as classical pTreg. In pTreg, CTLA-4 and PD-1 expression was also very low.

**Table 1 ijms-25-02031-t001:** Subjects enrolled in the study.

Subject Number	Weeds Gestation at Sample Collection	Age	Ethnicity	Natural Infection	Latest COVID-I9 Vaccine
1	10	35	Hispanic/White	N	12/2022, Moderna Bivalent
2	24	34	Non-hispanic/White	Y	1/2023, Moderna Bivalent
3	25	35	Non-hispanic/White	N	12/2022, Pfizer Bivalent
4	27	28	Non-hispanic/White	N	12/2022, Moderna Bivalent
5	19	39	Non-hispanic/White	Y	1/2023, Pfizer Bivalent
6	17	40	Non-hispanic/White	N	12/2022, Moderna Bivalent
7	26	36	Non-hispanic/White	N	12/2022, Pfizer Bivalent
8	17	31	Non-hispanic/White	N	1/2023, Pfizer Bivalent
9	20	31	Non-hispanic/White	N	2/2023, Pfizer Bivalent
10	23	28	Non-hispanic/White	Y	2/2023, Pfizer Bivalent
11	16	28	Non-hispanic/Cancasian and Asian	N	1/2023, Moderna Bivalent
12	26	34	Non-hispanic/White	N	2/2023, Moderna Bivalent
13	26	42	Non-hispanic/White	N	4/2023, Pfizer Bivalent
14	23	30	Non-hispanicWhite	N	5/2023, Moderna Bivalent
15	34	32	Non-hispanic/White	N	8/2023, Pfizer Bivalent
16	24	32	Non-Hispanic/Asian	Y	9/2023, Pfizer Bivalent
17	23	37	Non-hispanic/White	Y	9/2023, Pfizer Bivalent
18	21	34	Non-hispanic/White	Y	10/2023, Moderna
19	8	40	Non-hispanic/White	Y	9/2023, Moderna

## Data Availability

The authors confirm that the data supporting the findings of this study are available within the article.

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
