# Peer review of "T Cell Responses in Pregnant Women Who Received mRNA-Based Vaccination to Prevent COVID-19 Revealed Unknown Exposure to the Natural Infection and Numerous SARS-CoV-2-Specific CD4- CD8- Double Negative T Cells and Regulatory T Cells"

_ijms, 2024, doi:10.3390/ijms25042031_

Round 1
Reviewer 1 Report
Comments and Suggestions for Authors
In this study, Chambers et al. describe that most pregnant women, vaccinated with a mRNA based, spike-specific, Sars-CoV-2 vaccine show an efficient CD4+, CD8+ T cell response to spike (and occasionally to non-spike antigens), irrespective to prior natural infections, a finding also reported previously for normal individuals of both sexes. More importantly, the study shows that a consistent percentage of the responding T cells belong to the DN or to the Treg cell subset. Although somewhat observational rather than mechanistic, the study is interesting since it provides insights on the immunological status in pregnancy leading to the identification of possible T cell biomarkers to monitor the immune response in this condition. Based on these considerations, the paper certainly deserves to be published in IJMS.
However, there are a few points of major concern:
1. What is the percentage of spike- and non-spike-specific DN T cells or of Treg cells in vaccinated non-pregnant women and of healthy men? Is the proportion of these cells significantly higher in pregnancy?. The authors probably have these data already from previous studies: a direct comparison should be shown here to provide more convincing evidence to the readers.
2.Do DN T cells have a similar or the lower affinity/avidity for the spike peptides compared to CD4+ and CD8+ T cells? This issue can be explored by performing dose response curves with the different, purified T cell populations. If this is not feasible, the issue should be a point of discussion, together with some speculation on the cross-reactivities leading to the expansion of lower affinity/avidity T cells.
3. Do freshly isolated Treg cells have suppressor functions? Does this function become more powerful when the cells respond to spike and non-spike peptides? Treg suppressor function was nicely shown by the authors in a previous Autoimmunity paper. Clearly, these experiments are more difficult in the present setting; however, the authors should at least bring about the issue in the Discussion section, giving their ideas and pointing out the possible experimental limitations in this setting.
4. The isssue of the antigen-specificity of Treg cells requires further meditation. Evidence is given that Treg cells are activated and expand when total (unfractionated) PBMCs are stimulated with specific antigens in vitro . This is interpreted as an antigen specific Treg cell response, although an alternative explanation could be that T reg cells (or a proportion of them) proliferate in a by-stander manner owing to their well know capacity of capturing and of responding to IL2 released by other T cells concomitantly responding to the stimulating viral peptides in the same culture. The authors should bring about the issue, which deserves discussion if not amenable to experimental testing in the present setting.
There are additional minor concerns that should be amended:
1. The text is somewhat verbose and repetitive. For example, following the heading Results, there are a few lines of methodological details which are repeated immediately after the first sub-heading. The text could be shortened, avoiding repeating technical/methodological details that should be placed in the M.M section only. Furthermore, the authors should abstain from placing general considerations or even conclusions in the Results section; these would go more appropriately in the Discussion section.
2. The same critics apply to the Figure legends, which are in general too long and contain an excess of useless technical details making the reading difficult and preventing the full understanding of the general lay-out of the experiments. Furthermore, in the Figures, where the response to spike and non-spike peptides is shown, there is a horizontal dotted line, which may represent the threshold discriminating positive from negative results. This is not specified in legends and doubts are raised by knowing from the text that the authors apparently do not use the threshold indicated by the dotted lines to classify responders and non responders individuals, since the numbers given in text do not match those of responders and non-responders individuals provided by the Figure.
3 Fig 2 top. The anti CD3/anti CD28 annotation should be followed by "stimulation"
4. the Discussion section should be re-written to accomodate the issues outlined above and the considerations placed in the results section.
Comments on the Quality of English Language
The language is quite good, but the text requires modifications to avoid repetitions and useless technical details placed in inappropriate sections.
Author Response
In this study, Chambers et al. describe that most pregnant women, vaccinated with a mRNA based, spike-specific, Sars-CoV-2 vaccine show an efficient CD4+, CD8+ T cell response to spike (and occasionally to non-spike antigens), irrespective to prior natural infections, a finding also reported previously for normal individuals of both sexes. More importantly, the study shows that a consistent percentage of the responding T cells belong to the DN or to the Treg cell subset. Although somewhat observational rather than mechanistic, the study is interesting since it provides insights on the immunological status in pregnancy leading to the identification of possible T cell biomarkers to monitor the immune response in this condition. Based on these considerations, the paper certainly deserves to be published in IJMS.
We really appreciate the kind wording and interest in our work.
However, there are a few points of major concern:
- What is the percentage of spike- and non-spike-specific DN T cells or of Treg cells in vaccinated non-pregnant women and of healthy men? Is the proportion of these cells significantly higher in pregnancy?. The authors probably have these data already from previous studies: a direct comparison should be shown here to provide more convincing evidence to the readers.
Thank you for the opportunity to analyze DN T cells in the non-pregnant, vaccinated cohort of men and women, previously reported for the characterization of the SARS-CoV-2 spike-specific Treg population. DN T cells were numerous also in healthy vaccine recipients (Supplementary Figure 2), therefore not unique in pregnancy. We believe that the functional maturity of DN T cells in humans is an important finding that deserves to be reported.
2.Do DN T cells have a similar or the lower affinity/avidity for the spike peptides compared to CD4+ and CD8+ T cells? This issue can be explored by performing dose response curves with the different, purified T cell populations. If this is not feasible, the issue should be a point of discussion, together with some speculation on the cross-reactivities leading to the expansion of lower affinity/avidity T cells.
Our study involved testing numerous peptides in pools: a dose response would not be informative without narrowing down the immunodominant epitopes (out of the scope of this paper). The discussion has been carefully revised to address the reviewer’s concerns. We believe that the DN T cells that respond to antigens with CD8 markers are the results of overstimulation due to too frequent vaccine boosts with a narrow time in between injections, and the co-immunization with other vaccine formulations (i.e. Flu, RSV, etc). Thank you for the opportunity to improve the discussion, that is now revised and extended.
- Do freshly isolated Treg cells have suppressor functions? Does this function become more powerful when the cells respond to spike and non-spike peptides? Treg suppressor function was nicely shown by the authors in a previous Autoimmunity paper. Clearly, these experiments are more difficult in the present setting; however, the authors should at least bring about the issue in the Discussion section, giving their ideas and pointing out the possible experimental limitations in this setting.
Treg secrete IL-10 only when stimulated as shown in Figure 4. We apologize for the lack of formal suppression experiments in co-cultures due to cell limitations. Yes, the suppressor functions were depending upon TcR engagement. A paragraph has been added in the discussion within the limitations of the study.
- The isssue of the antigen-specificity of Treg cells requires further meditation. Evidence is given that Treg cells are activated and expand when total (unfractionated) PBMCs are stimulated with specific antigens in vitro . This is interpreted as an antigen specific Treg cell response, although an alternative explanation could be that T reg cells (or a proportion of them) proliferate in a by-stander manner owing to their well know capacity of capturing and of responding to IL2 released by other T cells concomitantly responding to the stimulating viral peptides in the same culture. The authors should bring about the issue, which deserves discussion if not amenable to experimental testing in the present setting.
We believe that the timing of the assay after stimulation is too short for a by-standard activation due to the IL-2 in the environment. In support of the SARS-CoV-2 specific Treg response, the percentage of Treg 48 hours after stimulation with SARS-CoV-2 peptides, in the presence or absence of IL-2 (Figure 5 panel B,) did not change. The percentage of antigen-specific Treg did not significantly change after a longer stimulation (Figure 5, panel C), waving concerns about a by-standard Treg expansion due to IL-2 secreted by pro-inflammatory, antigen-specific T cells.
There are additional minor concerns that should be amended:
- The text is somewhat verbose and repetitive. For example, following the heading Results, there are a few lines of methodological details which are repeated immediately after the first sub-heading. The text could be shortened, avoiding repeating technical/methodological details that should be placed in the M.M section only. Furthermore, the authors should abstain from placing general considerations or even conclusions in the Results section; these would go more appropriately in the Discussion section.
Thank you for the suggestions: we worked extensively in the revision. To repeat the markers used in the AIM assay, as well as the phenotype of different memory T cells lineages, at the beginning of the Results section, help readers not familiar, as the reviewer, with T cell recognition.
- The same critics apply to the Figure legends, which are in general too long and contain an excess of useless technical details making the reading difficult and preventing the full understanding of the general lay-out of the experiments. Furthermore, in the Figures, where the response to spike and non-spike peptides is shown, there is a horizontal dotted line, which may represent the threshold discriminating positive from negative results. This is not specified in legends and doubts are raised by knowing from the text that the authors apparently do not use the threshold indicated by the dotted lines to classify responders and non responders individuals, since the numbers given in text do not match those of responders and non-responders individuals provided by the Figure.
Thank you for giving us the opportunity to clarify this important point: we now explain the cut off for the SI (2), that defines a good T cell response. However, a T cell response significantly above the un-stimulated control cannot be ignored.
3 Fig 2 top. The anti CD3/anti CD28 annotation should be followed by "stimulation"
Thank you: the annotation in the Figure has been revised.
- the Discussion section should be re-written to accomodate the issues outlined above and the considerations placed in the results section.
We hope that the revised Discussion addressed all the reviewer’s concerns.
Reviewer 2 Report
Comments and Suggestions for Authors
In this this study, the authors conducted cohort study, in which pregnant who had or had not been vaccinated for COVID-19 were included. After stimulation, the authors study T cell activation, CCR6 expression, and T cell memory phenotypes. The manuscript and results are important; however, the manuscript needs major changes.
Major comments
1- The abstract should be modified, a good abstract should be subdivided in 4 part: Background, method, results and conclusion.
2- In the introduction, describe and give more information about SARS-CoV-2 infection and Covid-19 Pandemic in pregnant women
3- In the introduction, must be careful and distinguish between Covid-19 pandemic and the virus SARS-CoV-2 which cause the pandemic. Therefore, rephrase the sentence “In COVID-19, SARS-CoV-2 spike-specific regulatory T cells (Treg)…. the disease”.
4- In the introduction, describe the most important humoral, cellular immune response and its memory in vaccine recipients.
5- In the result part, the sentence “A total of 19 pregnant women were enrolled in the study and provided blood samples. The women ranged in age from 28 to 42 and provided blood samples at gestational weeks ranging from 10 to 34. Of the 19 subjects, 7 reported being aware that they had a previous COVID-19 infection” should be removed and placed in the material and methods.
6- In the material and method, give the number for the Ethics approval
7- In the part Study population, the author should give more information about the number of participants, exclusion and inclusion criteria, age, gestational weeks, previous infection or not…..etc.
8- When the blood samples were approved, where the women at the same period of pregnancy or different? If different how could you avoid the discrepancy between the women at different periods? This could influence significantly your results.
9- For the results, it would be more interesting, if you make diagram with different groups, pregnant with SARS-Cov2 infection and pregnant without previous infection, and in each group show the unstimulated and stimulated.
Minor comments
1- In the results part, the sentence “PBMC were stimulated in vitro for 24 hours with SARS-CoV-2 spike and non-spike peptides and T cell responses measured by flow cytometry using an Activation Induced Marker (AIM) assay…. to peptides divided by the unstimulated control signal” should be in the material and methods.
2- For the software, kits, antibodies and other reagent, mention the kit name, the clone for the antibody, company, city and country.
3- Statistical analysis should be the last part in material and methods.
4- The parts in material and methods and results must be numbered
5- Figure legend are very long, this should be shortened.
Comments on the Quality of English LanguageModerate editing of English language required
Author Response
Major comments
- The abstract should be modified, a good abstract should be subdivided in 4 part: Background, method, results and conclusion.
The abstract has been re-written as a structured abstract, as requested by the reviewer.
- In the introduction, describe and give more information about SARS-CoV-2 infection and Covid-19 Pandemic in pregnant women
We did add references in the introduction. We also emphasized that the scientific community is still missing several information on the basic immunology on SARS-CoV-2 infection in pregnancy.
- In the introduction, must be careful and distinguish between Covid-19 pandemic and the virus SARS-CoV-2 which cause the pandemic. Therefore, rephrase the sentence “In COVID-19, SARS-CoV-2 spike-specific regulatory T cells (Treg)…. the disease”.
Thank you for the careful editing: the change has been made.
- In the introduction, describe the most important humoral, cellular immune response and its memory in vaccine recipients.
We quoted two very comprehensive reviews: unfortunately, still little is known on the fine specificities of antibodies (especially T cell-dependent), their affinity and avidity and the relevance depending upon specificities in the viral clearance. The T cell responses have been studied mainly by our collaborator, Dr. Alex Sette: work in progress will narrow down the most relevant peptide epitopes and rank the immunodominance.
- In the result part, the sentence “A total of 19 pregnant women were enrolled in the study and provided blood samples. The women ranged in age from 28 to 42 and provided blood samples at gestational weeks ranging from 10 to 34. Of the 19 subjects, 7 reported being aware that they had a previous COVID-19 infection” should be removed and placed in the material and methods.
Thank you for the suggestion: the sentence is now in the material and methods.
- In the material and method, give the number for the Ethics approval
The IRB number is now reported.
- In the part Study population, the author should give more information about the number of participants, exclusion and inclusion criteria, age, gestational weeks, previous infection or not…..etc.
Please find the relevant information within the material and methods in the section “study population”.
- When the blood samples were approved, where the women at the same period of pregnancy or different? If different how could you avoid the discrepancy between the women at different periods? This could influence significantly your results.
Blood samples were drawn when the consent was signed by participants.
We apologize if were unclear in explaining the scope of the work: we enrolled in purpose women at different gestational weeks to address possible differences within the response to vaccination. All the subjects received 3 vaccine injections and were studied at similar times after vaccination (Table 1). The results suggested that different gestational weeks in pregnancy do not play a role in the potency of the vaccine.
For the results, it would be more interesting, if you make diagram with different groups, pregnant with SARS-Cov2 infection and pregnant without previous infection, and in each group show the unstimulated and stimulated.
The un-stimulated controls for all the T cell lineages are now provided in Supplementary Figure 1. Subjects that had a previous COVID-19 infection are indicated in every Figure and
color coded in red in Figure 1, panel B, that reports T cell memory.
Minor comments
- In the results part, the sentence “PBMC were stimulated in vitro for 24 hours with SARS-CoV-2 spike and non-spike peptides and T cell responses measured by flow cytometry using an Activation Induced Marker (AIM) assay…. to peptides divided by the unstimulated control signal” should be in the material and methods.
We respectfully point the reviewer attention that for readers not expert in human T cell recognition, these few lines are helpful to understand the experimental strategy.
- For the software, kits, antibodies and other reagent, mention the kit name, the clone for the antibody, company, city and country.
All the info related to the reagents are listed in the methods section.
- Statistical analysis should be the last part in material and methods.
Thank you for the suggestion.
- The parts in material and methods and results must be numbered
We respectfully point the reviewer attention that we followed the editorial instructions.
- Figure legend are very long, this should be shortened.
The Figure legends have been carefully revised.
Round 2
Reviewer 1 Report
Comments and Suggestions for Authors
The authors have clarified the points of concern and amended the manuscript, where necessary. Therefore, the paper can now be published in the present form.
Comments on the Quality of English LanguageThe quality of English is acceptable.
Author Response
Thank you for your kind decision.
Reviewer 2 Report
Comments and Suggestions for Authors
The author modified the abstract and addressed to most suggestions. However, there are few comments.
For some changes, the author doesn’t mention where he makes the modifications (Line Nr- Line Nr), which make it difficult for the reviewer to found where the modifications were done.
After addition of new references, the author should mention the number of the new references added. The reviewer cannot check all reference in the old version and new version to found which one is new.
In material and methods, exclusion and inclusion criteria were not mentioned
The author don’t understand my suggestion, make diagram with different groups means not every single patient, but a group of patients (group1: pregnant with SARS-Cov2 infection unstimulated, group2: pregnant with SARS-Cov2 infection stimulated, group3: pregnant without previous infection unstimulated, and group4: pregnant without previous infection stimulated).
For reagents and kits, the author doesn’t add city and/or country from where they got it.
Comments on the Quality of English Language
The manuscript needs editing service.
Author Response
The author modified the abstract and addressed to most suggestions. However, there are few comments.
We thank the reviewer for the kind comments on our revised abstract and manuscript.
For some changes, the author doesn’t mention where he makes the modifications (Line Nr- Line Nr), which make it difficult for the reviewer to found where the modifications were done.
All changes in the manuscript are marked in yellow.
After addition of new references, the author should mention the number of the new references added. The reviewer cannot check all reference in the old version and new version to found which one is new.
References too, are highlighted in yellow. Two comprehensive review on the cellular and humoral response to vaccination, and COVID-19 infection in pregnancy that reference all the appropriate papers have been added (ref 17 and 18).
In material and methods, exclusion and inclusion criteria were not mentioned.
The study population is well described in the methods and in Table 1. The criteria for inclusion are properly defined. Excluded were only women with pre-existing conditions. We added a sentence within the study population.
The author don’t understand my suggestion, make diagram with different groups means not every single patient, but a group of patients (group1: pregnant with SARS-Cov2 infection unstimulated, group2: pregnant with SARS-Cov2 infection stimulated, group3: pregnant without previous infection unstimulated, and group4: pregnant without previous infection stimulated).
A new Supplementary Figure 1 shows the four groups requested by the reviewer: however, in our study, several subjects were found to have contracted asymptomatic COVID-19, that we detected because of the non-spike-specific T cell response, therefore the “non-infection group” contains subjects that actually had the infection. We leave the Figure as supplementary because it could be very confusing for the reader.
For reagents and kits, the author doesn’t add city and/or country from where they got it.
The info has been added, as requested by the reviewer.